# Sustainability of Hotel, How Does Perceived Corporate Social Responsibility Influence Employees' Behaviors?

**Haiyan Kong [1,2], Naipeng (Tom) Bu [1,2,\*], Yue Yuan [1,2], Kangping Wang [1,2] and YoungHee Ro [3]**

[1] Business School, Shandong University, Weihai 264209, China; konghaiyan@sdu.edu.cn (H.K.); yuanyue0424@163.com (Y.Y.); kangpingwang331@163.com (K.W.)
[2] International Institute of Tourism Science, Shandong University, Weihai 264209, China
[3] School of Conflict Analysis & Resolution, George Mason University Korea, Inchon 21985, Korea; ceodrro@gmail.com
\* Correspondence: bunp@sdu.edu.cn

**Abstract:** This study sought to explore the influence of perceived corporate social responsibility (CSR) on organizational identification and organizational commitment. Based on extensive literature review, the theoretical model was proposed. Perceived CSR was developed as the determinant, organizational identification as the mediator, and organizational commitment as the outcome. Targeting highly educated employees, this study surveyed employees with bachelor degrees or above. A total of 836 data were collected, and structural equation modeling was analyzed with statistical software AMOS 21.0 software. The findings indicated that perceived CSR contributed positively to organizational identification and commitment. Apart from the direct positive effect on organizational commitment, organizational identification also mediated the relationship between perceived CSR and employee loyalty. The study highlighted the importance of fulfilling social responsibilities, encouraged managers to understand young well-educated employees from different perspectives, and also shed light on performing effective human resource management (HRM) activities, which can meet the UN Sustainable Development Goals and accelerate the related development in tourism and hospitality.

**Keywords:** perceived corporate social responsibility; organizational identification; organizational commitment; employees' behaviors; China

## 1. Introduction

As a basic unit of society, enterprises should evaluate the impact of their own activities on society when engaging in economic activities. Only when enterprises shoulder the social responsibility of helping countries become prosperous and strong and making people happy can enterprises achieve long-term and sustainable development. Therefore, increasing attention is being paid to how the perceptions of corporate social responsibility (CSR) influence employees' behaviors [1]. Although many companies are aware of the importance of corporate social responsibility, they still mainly focus on profits without considering sustainable development [2].

Competition for talents is increasingly fierce in the knowledge economy era. Thus, improving employee loyalty and maintaining effective competitiveness have become important concerns for enterprises [3]. With the improvement of the market economy, the work values of Chinese employees have also undergone great changes, especially the well-educated ones who no longer solely focus on economic benefits. Their subjective needs are increasingly diversified and complicated, and their work motivation and loyalty are strongly affected by CSR. For example, a survey indicated that employees perceived demonstrating a commitment to the environment was very important and required the

immediate attention of hotels [4]. Therefore, CSR is not only a social issue but also a need for the sustainable development of enterprises.

In terms of the research relating to perceived CSR, researchers have explored the policies, practices, and standards of CSR and sustainability in the hospitality industry [5,6]. Previous studies have explored the determinants and outcomes of perceived CSR. For instance, it has been found that National (not natural) culture has an effect on CSR in the tourism and hospitality industry [7]. The mediating role of collective self-esteem in the relationship between employees' perceived CSR and their work commitment has been examined [8]. Perceived CSR is an important predictor of customer satisfaction and hotel profit [9]. And Farooq et al. [10] examined the direct effect of employees' perceived CSR on affective organizational commitment. However, a very small number of studies have been conducted to explore the relationship between employees' perception of CSR and their organizational behaviors.

This research, thus, aims to investigate the relationship between perceived CSR and employee loyalty. It proposes the perceived CSR as the determinant, while organizational identification and employees' organizational commitment are the outcomes. This study also aims to explore the mediating effect on the relationships between perceived CSR and organizational commitment.

## 2. Theoretical Basis and Concept Definition

### 2.1. Social Identity Theory

Social identity refers to individuals' cognition of their belonging to a certain group and the emotion and value brought by that [11]. Social identity theory explains its behavioral relationship with a group that can be used to determine how employees identify with an organization and how this identification affects their attitudes and behaviors. By comparing the advantages and disadvantages of internal and external groups, employees can develop prejudice against external groups and preference integration for internal groups, thus obtaining positive social identity [12]. Only when enterprises establish social responsibility and apply it to specific activities by making employees feel this responsibility can employees trust and recognize such enterprises, consciously protect enterprise interest, and serve and promote enterprise development.

On the basis of such an analysis, this study obtains employees' perceived CSR as the influencing factor of organizational identity and employee loyalty. To explore the relationship among the three factors, organizational identity is also regarded as the influencing factor of employee loyalty.

### 2.2. Corporate Social Responsibility (CSR)

Corporate social responsibility initially developed from the pure pursuit of the economy to care about the narrow sense of social responsibility, which is beyond economy and legislation. CSR then develops the broad sense of social responsibility, which requires enterprises to undertake additional responsibilities that are beneficial to social welfare, such as ethics and charity, from the social perspective. CSR is subsequently subdivided into social responsibilities for each stakeholder, not solely to maximize economic profit, but also to represent humanitarian social causes [13]. With the deepening of research, the connotation of CSR is increasingly rich and specific, but its definition remains unrecognized because scholars understand and explore CSR from different research perspectives. Given that employees are one of the important stakeholders of enterprises, this study focuses on the relationship between organizations and employees. Considering the practical and scientific knowledge of CSR from the perspective of stakeholders, this research agrees with scholars' definition of CSR from the perspective of stakeholders. Therefore, CSR refers to enterprises' responsibility to earn profits for shareholders and take social responsibility for other stakeholders, such as employees, society, and the environment. Social responsibility involves the observation of business ethics, production safety, and occupational health; the protection of laborers' legal rights and interests and environment; and resource saving [14]. The existing research proposes an integrated model incorporating the interrelationships among CSR

practices, organizational culture, and company reputation to improve firm performance in the hotel and tourism business [15].

CSR has a positive impact on employees' behaviors by the serial mediation of organizational pride and identification [16]. With an increasing concern for corporate social responsibility (CSR), leading companies in diverse industries, driven by companies' stakeholders, customers, societies, and governments, are hastening initiatives to reveal their CSR commitments [17]. A social entrepreneur is an individual that generates a company to create social value. Social entrepreneurs tend to develop these initiatives because they have a strong social direction. Any business plan that is developed with a social objective to diminish a social problem and to make social value can be regarded as a social enterprise [18].

### 2.3. Organizational Identification

From the perspective of emotional motivation, organizational identity refers to the sense of dependence and belonging when employees identify with the values, codes of conduct, and management concepts of an enterprise [19]. And it also refers to how an employee relates to the organization [20]. Organizational identification is related to the theory of social identity—it means that when a person identifies strongly in the group, then he/she will participate strongly. Organizational identification affects one's job crafting and adaptability [21] and is highly related to his/her workplace behavior [22]. Furthermore, employees' job insecurity and job performance are affected by organizational identification [23].

### 2.4. Organizational Commitment

Since organizational commitment (OC) is a psychological state that binds the individual to the organization, the nature of that psychological state should be recognized. There are different views about the description of that psychological state. Attempts to refine these differences have resulted in different multidimensional conceptualizations for OC, and many academics have accepted OC as a multidimensional construct [24]. What differentiates the extents of OC in multidimensional conceptualizations is the fundamental psychological state convincing the employee toward a course of action [24]. However, there is no agreement on the dimensionality of OC. There are different models trying to explain the dimensions of OC. Angle and Perry [25] suggested a two-dimensional framework: value commitment and commitment to stay. Organizational commitment reflects the mental state between employee and organization, explaining employees' decisions of whether to remain with the organization [26]. It includes three components, namely an affective component, continuance component, and normative component. Emotional commitment reflects commitment based on emotional ties the employee grows with the organization primarily through positive work experiences. Normative commitment reflects commitment based on perceived responsibility towards the organization, for example, embedded in the norms of reciprocity. Persistence commitment reflects commitment based on the perceived costs, both economic and social, of leaving the organization [24].

## 3. Literature Analysis and Research Hypothesis

This study aims to explore the relationships among employees' perceived CSR, organizational identity, and employee loyalty. On the basis of the given analysis, this research sets employees' perceived CSR as an independent variable and organizational identity and employee loyalty as dependent variables. Therefore, the mediating effect of organizational identity can be measured.

### 3.1. Relationship between Perceived CSR and Organizational Identity

Employees have high organizational identification with enterprises that actively undertake social responsibility. Otherwise, employees have low identification [27]. Given that employees' perception is influenced by interpersonal relationships and moral accomplishments and considering the psychological sense of fairness, the CSR performance of various stakeholders' behaviors can

positively affect employees, improve their organizational commitment, and recognize the value of self-realization [28]. According to Davis [29], "the firm's consideration of and reply to issues beyond the narrow economic, legal requests of the firm to achieve the social benefits along with the conventional economic profits which the firm seeks". Ishikawa [30] asserts this statement while describing CSR: "the first concern of a company is the happiness of the people connected to it. If the people do not feel happy, that company does not deserve to exist". In a related vein, Jacques [31] states that "from many people, the marvelous appeal of quality is the opportunity to do good, to increase the workplace, to foster standards of living, and to achieve excellence". Disclosing the social obligation may provide benefits to the company in a form of improved reputation and market value [32].

Moir [33], and Idowu and Towler [34] state that CSR can produce supportive communities, increased customer loyalty, improved quality and productivity, and greater employee loyalty and relation. "Strategic CSR", advocated by Porter and Kramer [35], aims to employ CSR as a means to promote competitive advantage. Chinese scholars have investigated the relationship between CSR and organizational identity and found that the better the CSR performance, the higher the level of employees' identification with an organization [36,37]. If enterprises perform their CSR well, it may affect employees' organizational behaviors, such as organizational identity. Thus, the following is proposed:

**Hypothesis 1 (H1).** *Employees' perceived CSR positively relates to organizational identity.*

**Hypothesis 1a (H1a).** *Perceived employee dimension of CSR positively relates to organizational identity.*

**Hypothesis 1b (H1b).** *Perceived customer dimension of CSR positively relates to organizational identity.*

**Hypothesis 1c (H1c).** *Perceived integrity dimension of CSR positively relates to organizational identity.*

**Hypothesis 1d (H1d).** *Perceived charity dimension of CSR positively relates to organizational identity.*

**Hypothesis 1e (H1e).** *Perceived environment dimension of CSR positively relates to organizational identity.*

*3.2. Relationship between Perceived CSR and Organizational Commitment*

From the perspective of human resource development, enterprises' behaviors of assuming social responsibility may play a positive role in improving employees' recognition, loyalty, and satisfaction [38], and thus stimulate human resource development and attract high-quality talents [39]. Companies contain the language of CSR and take measures to reform the management system to make them more responsive to the environmental and social concerns of different stakeholders [40]. Further, Mostovicz [41] states that "if the goal of the corporations is only structured towards profit maximization, then the human values fade away, enslaving corporate employees to the target of the organization as a whole". From the perspective of resource theory, CSR behavior can establish corporate image and improve corporate reputation, thus influencing employees' attitudes toward enterprises and creating internal value [42]. When discussing social responsibility and international competitiveness, Chinese scholars suggest that CSR contributes good social image, obtains employee satisfaction, and improves corporate reputation. CSR aids the establishment of good relationships between customers and employees; boosts employees' morale, work satisfaction, and sense of belonging; and stimulates their commitment and loyalty, thereby advancing production efficiency [43] and reducing the employee turnover rate [44]. Therefore, the following is suggested:

**Hypothesis 2 (H2).** *Employees' perceived CSR positively relates to organizational commitment.*

**Hypothesis 2a (H2a).** *Perceived employee dimension of CSR positively relates to organizational commitment.*

**Hypothesis 2b (H2b).** *Perceived customer dimension of CSR positively relates to organizational commitment.*

**Hypothesis 2c (H2c).** *Perceived integrity dimension of CSR positively relates to organizational commitment.*

**Hypothesis 2d (H2d).** *Perceived charity dimension of CSR positively relates to organizational commitment.*

**Hypothesis 2e (H2e).** *Perceived environment dimension of CSR positively relates to organizational commitment.*

*3.3. Relationship between Organizational Identity and Commitment*

The concept of organizational commitment (OC) and identity has been an important theme in organizational behavior research. Organizational commitment and job satisfaction are considered precedents of employee performance [45]. Bergami and Bagozzi [46] explored the relationship between organizational identity and affective commitment and found that organizational identity is positively correlated with affective commitment. Another significantly positive correlation exists between the two dimensions of organizational identity, including attributive identity, successful identity, and emotional loyalty [47]. The most commonly investigated type of OC is attitudinal, which was developed by Mowday and his colleagues [48]. They outline attitudinal OC as the strength of an individual's identification with an organization and its goals and values. The higher the degree of employees' identification with an organization, the more they show a supportive attitude and cooperative behavior toward the organization [20]. In addition, employees with high recognition more likely accept organizational requirements and make personal decisions conducive to the realization of organizational goals [49].

Therefore, strong organizational commitment can cause an inevitable change in employee psychology, attitude, and behavior, all of which can upgrade an employee's self-concept to the organization level, cause him/her to regard himself/herself as an indispensable part of an organization, make himself/herself actively concerned, and can result in a strong sense of responsibility and belonging, thus improving employee loyalty [50]. This analysis leads to the third hypothesis:

**Hypothesis 3 (H3).** *Organizational identity positively relates to organizational commitment.*

*3.4. Mediating Effect of Organizational Identity*

This study defines organizational identity as a cognitive psychology of employees. Organizational identity is a process from employees' cognition to their emotional dependence and commitment. The stronger the degree of employees' organizational identification, the stronger the positive correlation between their perceived CSR and continuous commitment [51]. Moreover, the mediating effect of organizational identity on CSR and employee loyalty has different impacts on different dimensions of employee loyalty. The influence on affective and normative commitment is in complete mediation, whereas that on continuous commitment is in partial mediation. That is, CSR can directly affect continuous commitment [52].

By perceiving CSR, employees can improve their identification with an organization, develop a sense of trust and belonging, and achieve loyalty. Therefore, the following assumption is made:

**Hypothesis 4 (H4).** *Organizational identity plays a mediating role between perceived CSR and organizational commitment.*

According to the analysis above, Figure 1 depicts the theoretical model of the study.

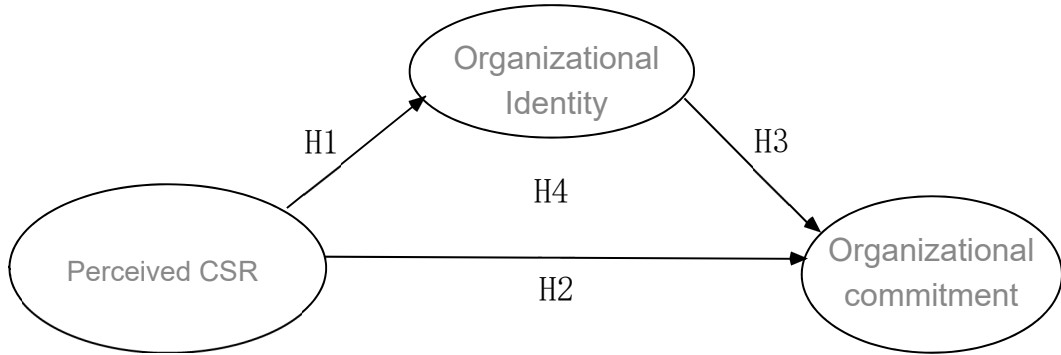

**Figure 1.** Theoretical model.

## 4. Research Methodology

### 4.1. Research Design

We used the convenience sampling method in our study. The target population of this study was the well-educated employees in the service industry. The participants were selected according to the following conditions: should have a bachelor degree or above and have worked for more than one year. The second condition is valid because interviewees cannot comprehensively understand an organization's values, codes of conduct, and business philosophy if the entry time is too short, and thus may affect the reflection of the actual situation.

The questionnaire was divided into four parts. For the first three parts, the researchers investigated employees' perceived CSR, organizational identity, and emotional commitment, respectively. The fourth part was for background investigation. All measurement items in the questionnaire were scored according to a seven-point Likert scale (1 = strongly disagree, 7 = strongly agree).

### 4.2. Measurement

For CSR measurement, the subjective perception of employees was adopted, and all aspects of CSR were scored by employees. This research used the CSR scale revised by He et al. [53] that is based on Turker's [54] scale, which was also revised to follow Chinese conditions. The scale involved five aspects of CSR, including the responsibilities for employees, consumers/products, integrity and justice, charity, and environment, which were a total of 22 items. Sample items include "special measures taken by a company to reduce its negative impact on the environment".

Organizational identity employed the scale developed by Mael and Ashforth [20], which is concise, understandable, and distinguishable from employee loyalty. This scale is also consistent with the concept of this study. Sample questions include "I regard the success of the company as my success and the glory of the company as my glory".

Organizational commitment adopted the scale developed by Allen and Meyer [26], which is widely applied and has great responsibility and validity. Sample items include "The company gives me a sense of belonging" and "I would love to work for this company all the way".

### 4.3. Data Collection

First, a pilot survey was conducted to verify the validity and reliability of the scale. The reliability of the constructs and wordings was assessed, and 135 valid questionnaires were collected. After ensuring the rationality of the questionnaire, the main survey was conducted nationwide and targeted well-educated young employees. Data were collected in two ways—online survey and spot investigation. While conducting the online survey, questionnaires were distributed via Sojump, an online distribution platform. After completing a questionnaire, the participants received cash as an incentive. For spot investigations, the researchers contacted key persons in the service industry, such as human resource or hotel marketing managers. With the assistance of these key persons, questionnaires

were distributed and collected on the spot. A total of 900 questionnaires were collected. After removing invalid questionnaires, the number of valid ones was 836 with an effective rate of 92.89%.

## 4.4. Data Analysis Method

Data analysis was performed according to the following steps: (1) data screening and descriptive analysis, (2) reliability analysis, (3) individual measurement model test, (4) overall measurement model test, and (5) structural model test. While examining the individual measurement model, data were randomly divided into two equal parts: exploratory factor analysis (EFA) and confirmatory factor analysis (CFA). All data were used to examine the overall measurement and structural models. SPSS 20.0 software was employed for descriptive analysis, the reliability test, and EFA, whereas AMOS 21.0 statistical software was used for deterministic factor analysis and structural equation analysis.

## 5. Research Results

### 5.1. Descriptive Statistical Analysis of Samples

This study targeted highly educated employees. As shown in Table 1, about 60.2% of the participants had received bachelor degrees, and 39.8% of the participants had obtained even higher degrees, that is, master and PhD degrees. The second characteristic of the participants was that they were young talents. Most of the employees surveyed were aged below 25 years old, and 38.2% of them were aged between 16 and 35 years old.

**Table 1.** Demographic factors of respondents (*N* = 836).

| Item | Personality Characteristic | Frequency | Proportion (%) |
|---|---|---|---|
| Gender | Male | 380 | 45.5 |
| | Female | 456 | 54.5 |
| Age | 25 years or below | 452 | 54.1 |
| | 26–30 years | 262 | 31.3 |
| | 31–35 years | 58 | 6.9 |
| | 36–45 years | 31 | 3.7 |
| | Above 46 years | 33 | 3.9 |
| Education | Bachelor | 503 | 60.2 |
| | Master | 306 | 36.6 |
| | Doctor or above | 27 | 3.2 |
| Years of Working | 1–3 years | 652 | 78.0 |
| | 4–6 years | 82 | 9.8 |
| | 7–9 years | 35 | 4.2 |
| | 10 years or above | 67 | 8.0 |
| Years of Working in the Company | 1–3 years | 688 | 82.3 |
| | 4–6 years | 72 | 8.6 |
| | 7–9 years | 26 | 3.1 |
| | 10 years or above | 50 | 6.0 |
| Disposable Income per Month (Pre-Tax) (In terms of Chinese Yuan) | 5000 RMB or below | 289 | 34.6 |
| | 5001–10,000 RMB | 386 | 46.2 |
| | 10,001–15,000 RMB | 101 | 12.1 |
| | 15,001–20,000 RMB | 29 | 3.5 |
| | 20,000 RMB or above | 31 | 3.7 |

*5.2. Individual Measurement Model*

5.2.1. Perceived CSR Measurement Model

EFA of Perceived CSR

The result of Bartlett's test of sphericity was significant, and the Kaiser–Meyer–Olkin (KMO) measurement of sampling adequacy was 0.93, indicating that the correlation and stability of each measurement item were good and factor analysis shows a satisfying result. Five components with eigenvalues greater than one were extracted, namely, responsibility for employees, integrity and justice, charity and public welfare, consumers/products, and environment. The total variance explained was 70.01%. Table 2 shows that the overall structural reliability was 0.93, and the alpha values in the five dimensions ranged from 0.85 to 0.90, all of which were greater than 0.70. Thus, high internal consistency and stability existed in each scale item.

**Table 2.** Exploratory Factor Analysis results of perceived Corporate Social Responsibility ($N = 418$).

| Item | Fac. | E. V | V. E (%) | CITC | α after Deleting the Item | α |
|---|---|---|---|---|---|---|
| | | | | | | 0.93 |
| Factor 1: responsibility for employees | | 3.64 | 16.52 | | | 0.86 |
| CSR6 | 0.80 | | | 0.78 | 0.81 | |
| CSR1 | 0.74 | | | 0.71 | 0.82 | |
| CSR4 | 0.73 | | | 0.67 | 0.83 | |
| CSR3 | 0.70 | | | 0.58 | 0.85 | |
| CSR2 | 0.68 | | | 0.66 | 0.83 | |
| CSR5 | 0.60 | | | 0.51 | 0.86 | |
| Factor 2: responsibility for integrity and justice | | 3.26 | 14.84 | | | 0.90 |
| CSR14 | 0.76 | | | 0.78 | 0.87 | |
| CSR13 | 0.76 | | | 0.78 | 0.87 | |
| CSR15 | 0.76 | | | 0.78 | 0.87 | |
| CSR12 | 0.72 | | | 0.70 | 0.89 | |
| CSR11 | 0.63 | | | 0.70 | 0.89 | |
| Factor 3: responsibility for charity and public welfare | | 3.06 | 13.92 | | | 0.88 |
| CSR18 | 0.86 | | | 0.78 | 0.83 | |
| CSR17 | 0.83 | | | 0.77 | 0.84 | |
| CSR19 | 0.81 | | | 0.74 | 0.85 | |
| CSR16 | 0.74 | | | 0.68 | 0.87 | |
| Factor 4: responsibility for consumers/products | | 2.95 | 13.39 | | | 0.85 |
| CSR10 | 0.80 | | | 0.70 | 0.81 | |
| CSR9 | 0.79 | | | 0.70 | 0.81 | |
| CSR7 | 0.75 | | | 0.66 | 0.83 | |
| CSR8 | 0.72 | | | 0.72 | 0.80 | |
| Factor 5: responsibility for environment | | 2.50 | 11.34 | | | 0.87 |
| CSR21 | 0.83 | | | 0.76 | 0.80 | |
| CSR20 | 0.82 | | | 0.75 | 0.81 | |
| CSR22 | 0.79 | | | 0.72 | 0.84 | |

Fac. = Factor, E. V = Eigenvalue, V. E = Variance Explained (%), CITC = Corrected Item Total Correlation.

CFA of Perceived CSR

The construct of perceived CSR was a multi-level, latent variable structure, thus second-order CFA was conducted. The first-order CFA examined the relationship between factors and measurement items. Results of the AMOS analysis revealed that χ2 = 405.13, degree of freedom (df) = 199, comparative fit index (CFI) = 0.96, goodness-of-fit index (GFI) = 0.92, and root mean square error of approximation (RMSEA) = 0.05, thus confirming the good fit between the model and data. The standardized parameter estimate value of all perceived CSR items was greater than 0.5, and the Critical Ration (C.R.) value (t value) was greater than 1.96, suggesting that the convergent validity was satisfactory. Table 3 presents that the average variance extracted (AVE) of the five factors ranged from 0.54 to 0.69, which was above 0.5 and was greater than the square of the correlation coefficient. The reliability of the five dimensions met the requirement of being greater than 0.70. Therefore, perceived CSR obtained good convergent and discriminant validity.

**Table 3.** Correlations (squared correlation), Average Variance Extracted, and mean (*N* = 418).

| Item | Employee | Customer | Integrity | Charity | Environment |
|---|---|---|---|---|---|
| Employee | 1.00 | | | | |
| Customer | 0.54 ** (0.29) | 1.00 | | | |
| Integrity | 0.60 ** (0.36) | 0.58 ** (0.34) | 1.00 | | |
| Charity | 0.46 ** (0.21) | 0.41 ** (0.17) | 0.57 ** (0.32) | 1.00 | |
| Environment | 0.45 ** (0.20) | 0.53 ** (0.28) | 0.57 ** (0.32) | 0.54 ** (0.29) | 1.00 |
| Reliability | 0.88 | 0.83 | 0.89 | 0.90 | 0.87 |
| AVE | 0.54 | 0.55 | 0.62 | 0.68 | 0.69 |
| Mean | 5.09 | 5.13 | 4.54 | 4.70 | 4.78 |
| Standard deviation | 0.97 | 1.01 | 1.13 | 1.10 | 1.26 |

Notes: ** means *p* < 0.01.

The second-order CFA was then conducted. The goodness-of-fit index of the model was $\chi^2$ = 427.50, df = 204, CFI = 0.96, GFI = 0.92, and RMSEA = 0.05, indicating the model fit the sample data well. Standardized parameter estimates of employees, customer, integrity, charity, and environment responsibility were 0.76, 0.77, 0.87, 0.71, and 0.77, respectively. The t value (C.R.) ranged from 10.44 to 12.16. Thus, the measurement model of perceived CSR examination reached a satisfactory level.

5.2.2. Measurement Model of Organizational Identity

EFA of Organizational Identity

Six measurement items existed in organizational identity, and the explained variance was 67.08%. The result of Bartlett's test of sphericity was significant, and the KMO measurement of sampling adequacy was 0.91, suggesting that the correlation patterns were relatively compact and generated distinct and reliable factors. The overall reliability alpha value was 0.90, and the influence of deleting a certain item on the overall reliability was not high. The factor loading ranged from 0.69 to 0.75, which were all greater than 0.30. Thus, the measurement items were able to reflect organizational identity well, and the scale was valid and reliable.

CFA of Organizational Identity

The CFA results revealed that $\chi^2$ = 9.26, df = 9, CFI = 0.99, GFI = 0.99, and RMSEA = 0.01, suggesting the model fit well with the data. The standardized parameter estimates ranged from 0.76 to 0.82, and the values of C.R. (t value) ranged from 17.29 to 18.84, all above 1.96. The AVE value was

0.64, greater than 0.50 and the square of the correlation coefficient. The results combined suggest that satisfactory convergent and discriminant validity is achieved [55].

### 5.2.3. Measurement Model of Organizational Commitment

### EFA of Employee Loyalty

The EFA results of organizational commitment showed that Bartlett's test of sphericity was significant, and the KMO measurement of sampling adequacy was 0.94, indicating that the correlation patterns were relatively compact and generated distinct and reliable factors. Three dimensions with eigenvalues above one were extracted, namely, normative, continuance, and affective commitment. The explained total variance was 66.33%. The overall structural reliability was 0.93, and the alpha values of the three dimensions ranged from 0.84 to 0.93. Therefore, the measurement items were able to reflect the variable of organizational commitment well, and the scale was valid and reliable.

### CFA of Organizational Commitment

First- and second-round CFA were conducted for organizational commitment. The first-order CFA was run to test the relationship between factors and measurement items. The results showed that $\chi^2 = 240.91$, df = 132, CFI = 0.98, GFI = 0.94, and RMSEA = 0.04, indicating a good fit between the model and the data. The reliability of the three dimensions of organizational commitment was 0.85 (affective commitment), 0.91 (continuance commitment), and 0.93 (normative commitment), respectively. The AVE value of the three dimensions was 0.50, 0.63, and 0.70, respectively, above 0.50 and greater than the squared correlation coefficient. Hence, organizational commitment had good convergent and discriminant validity.

The second-order CFA yielded the model fit index as follows: $\chi^2 = 240.91$, df = 132, CFI = 0.98, GFI = 0.94, and RMSEA = 0.04, which means the model fit the data well. The standard factor loading ranged from 0.74 to 0.77, which were all above 0.5. The t values (C.T.) were 8.86 and 9.05, which were greater than 1.96, confirming that organizational commitment had satisfactory convergent and discriminant validity.

### 5.3. Overall Measurement Model Test

The overall measurement model was tested with all data, and the goodness-of-fit results were $\chi^2 = 218.35$, df = 74, CFI = 0.97, GFI = 0.97, and RMSEA = 0.05. As shown in Table 4, the reliability of each construct was 0.84, 0.91, and 0.77, respectively. The AVE values were all above 0.50, and greater than the square of the correlation coefficient, indicating that the overall measurement model was valid and reliable.

**Table 4.** Correlation (squared correlation), reliability, and mean of overall measurement model.

| Structure | CSR | OI | OC |
|---|---|---|---|
| Corporate Social Responsibility | 1.00 | | |
| Organizational Identity | 0.33 ** (0.11) | 1.00 | |
| Organizational Commitment | 0.30 ** (0.09) | 0.37 ** (0.14) | 1.00 |
| Reliability | 0.84 | 0.91 | 0.77 |
| AVE | 0.51 | 0.62 | 0.52 |
| Mean | 4.86 | 4.84 | 4.59 |
| Standard Deviation | 0.84 | 1.11 | 0.86 |

Notes: ** means *p* < 0.01.

### 5.4. Structural Model and Hypothesis Testing

The final structural model was tested with all data using the AMOS software package. The model fit indices were $\chi^2$ = 218.35, df = 74, CFI = 0.97, GFI = 0.97, and RMSEA = 0.05. Based on the CFI, GFI, and RMSEA values, the structural model can be considered to fit the sample data fairly well.

As perceived CSR includes five dimensions, namely, employees, consumers, integrity, charity, and environment, this study further examined the relationships among them and two other constructs. As shown in Table 5, the path coefficient value and significance level together indicate that the structural paths were both positive and significant, and thus that all of the direct positive relationships were supported.

**Table 5.** Path results for the final structural model (hypotheses testing).

| Hypotheses/Path | Coefficient | T Value | Results |
|---|---|---|---|
| H1: Perceived CSR –> Organizational identity | 0.37 | 8.87 *** | Supported |
| H1a: Employee –> Organizational identity | 0.33 | 8.45 *** | Supported |
| H1b: Customer –> Organizational identity | 0.22 | 5.44 *** | Supported |
| H1c: Integrity –> Organizational identity | 0.31 | 8.08 *** | Supported |
| H1d: Charity –> Organizational identity | 0.27 | 6.87 *** | Supported |
| H1e: Environment –> Organizational identity | 0.30 | 7.80 *** | Supported |
| H2: Perceived CSR –> Organizational commitment | 0.23 | 5.26 *** | Supported |
| H2a: Employee –> Organizational commitment | 0.25 | 6.00 *** | Supported |
| H2b: Customer –> Organizational commitment | 0.18 | 4.19 *** | Supported |
| H2c: Integrity –> Organizational commitment | 0.16 | 3.81 *** | Supported |
| H2d: Charity –> Organizational identity | 0.14 | 3.49 *** | Supported |
| H2e: Environment –> Organizational commitment | 0.10 | 2.38 ** | Supported |
| H3: Organizational identity –> Organizational commitment | 0.37 | 7.85 *** | Supported |
| H4: Mediating effect of organizational identity | 0.14 | 5.88 *** | Supported |

Note: ** means $p < 0.01$, *** means $p < 0.001$.

The mediating effect of organizational identity was examined according to the formulae of MacKinnon et al. [56] The indirect effect = a × b (a, indicating the path coefficient of the association between the exogenous variable and the mediator and b, the path coefficient of the association between the mediator and the outcome). The significance level was calculated using the Sobel test, which seems to perform best in a Monte Carlo study [56]. Table 5 shows the results of the structural model and all hypotheses testing. Figure 2 shows the final structural model with direct path results.

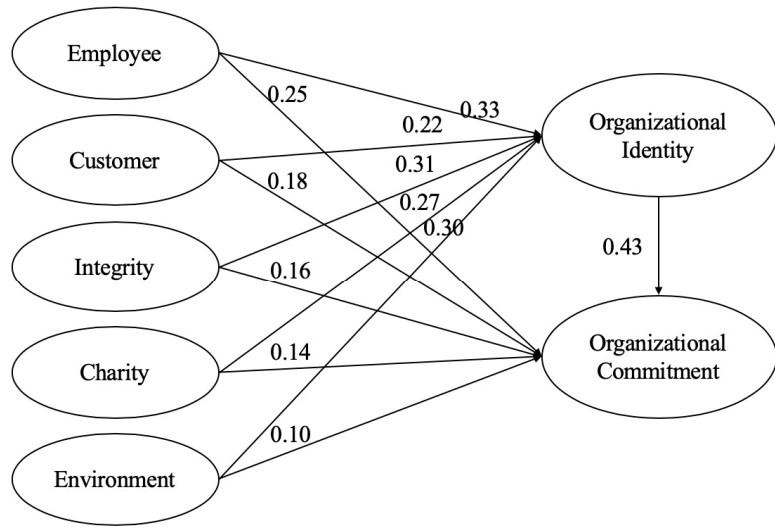

**Figure 2.** Final structural model.

## 6. Discussion

First of all, previous studies of CSR mostly focused on performance or corporate reputation. Few attentions have been paid to employees' perspectives. This research examined the effects of the five dimensions of perceived CSR on organizational identity and commitment. As a result, a theoretical model, which initially linked perceived CSR and employee behavior, was developed. It expanded to the exploration of the outcomes of perceived CSR.

Then, this study highlighted the importance of fulfilling social responsibilities. Although certain enterprises have realized this importance, not all of them have fulfilled such responsibilities in practice [57]. Due to economic benefits, certain enterprises thought investing to fulfill social responsibilities may increase operating costs, which may affect economic benefits and competence. The empirical results proved the importance of carrying out social responsibilities.

Finally, the five dimensions of perceived CSR (employees, customers, integrity and justice, charity, and environment) have significantly positive influences on organizational identity and commitment. HR managers must introduce the company strategies to employees and link organizational policies with employees' behaviors. For example, when companies actively perform responsibility for employees, consumers, integrity and justice, charity, and environment, their employees may feel that they are working for a responsible and mission-driven enterprise, and have a high level of job satisfaction [58]. The higher the degree of employees' perceived CSR performance, the more likely these employees are loyal to the enterprise.

## 7. Conclusions

Firstly, combined with extensive literature analysis, a theoretical model of employees' perceived CSR, organizational identity, and employee loyalty was proposed. Employees' perceived CSR has a significantly positive effect on organizational identity and organizational commitment, and organizational identity works as a mediator between perceived CSR and employee loyalty. In the model, perceived CSR was developed as the predictor, organizational identity as a mediator, and employee commitment as the outcome. And stronger micro-foundations of CSR research could be formed through understanding employees' emotional, attitudinal, and behavioral reactions to their perceptions of their employers' CSR. Due to the fact that research in this area is rare, the findings of this study may serve as the foundation for future research.

Secondly, the findings of this study indicated that when employees perceive their organizations involvement in CSR, they tend to generate a strong sense of belonging and emotional support for

their organization, treat themselves as part of the organization, and thus form a strong loyalty to the organization, which may enrich the knowledge of HRM and shed light on performing effective activities [59]. Thus, fulfilling social responsibility is not a short-term capital investment, but a long-term investment, which is beneficial to sustainable development.

Thirdly, the enterprises may actively fulfill their responsibilities for employees, consumers, integrity and justice, charity, and environment to enhance employees' sense of identity to an organization, leading to organizational commitment and employee loyalty. It encourages managers to understand young well-educated employees from different perspectives, and aligns their corporate values and interests with that of employees when determining CSR activities [13]. Generation Y employees are concerned about career development and work–life balance [60,61]. These findings may stimulate future studies to explore perceived CSR from different perspectives. Thus, providing cultural training and enhancing employee trust and recognition are important to make employees perceive and understand an enterprise's social responsibility. Effective HR activities and companies' social responsibilities may be combined for a win-win situation.

The research concludes by making some recommendations and suggestions of a practical character and highlighting future research directions.

## 8. Limitations and Suggestions for Future Research

One limitation was related to the convenience sampling method, which may impact the generalizability of the research. Future studies are suggested to adopt a quota sampling method for sample data collection, so the results may become representative and scientific.

Another limitation was about the examination of the organization commitment construct. The three dimensions of organizational commitment are well known. This study only examined the influence of perceived CSR on organizational commitment, not on the three dimensions of it. This may leave room for future studies.

**Author Contributions:** Conceptualization, H.K. and N.B.; methodology, Y.Y.; software, Y.Y. and K.W.; validation, H.K., N.B. and Y.R.; formal analysis, Y.Y.; investigation, K.W.; resources, Y.Y. and K.W.; data curation, Y.Y. and K.W.; writing—original draft preparation, N.B.; writing—review and editing, N.B. and H.K.; visualization, H.K. and N.B.; supervision, H.K.; project administration, H.K.; funding acquisition, H.K.

**Funding:** This research was funded by Youth Team Grant Fund of Shandong University (Weihai) and Business School. The culture image and competitiveness of tourism in China, grant number #2018WQTDXM003.

**Acknowledgments:** The authors would like to thank the support of Youth Team Grant Fund of Shandong University (Weihai) and Business School. The culture image and competitiveness of tourism in China. #2018WQTDXM003.

**Conflicts of Interest:** The authors declare no conflict of interest.

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
