# Peer review of "Sustainability of Hotel, How Does Perceived Corporate Social Responsibility Influence Employees’ Behaviors?"

_sustainability, doi:10.3390/su11247009_

Round 1

Reviewer 1 Report

Thank you for addressing the majority of my concerns.  However, you failed to address the issue of the sample and what make it a convivence sample.  One sentence in the research design section would be enough.

The new citation in section 2.2 does not have a year and please clarify if the income in Table one is monthly or not. 

Reviewer 2 Report

The research is relevant, topic is very contemporary, theoretical background comprehensive, barriers explained. Hypothesis is rather less ambitious, but elaborated into details. Theoretical sources as well as application is related to the regional context, which supports the application value. I recommend to reconsider the perspective of happiness of people in companies, which has appeared in the paper.  The paper is valuable contribution to the research of CSR in the region. There are several typing errors.  

Author Response

This manuscript is a resubmission of an earlier submission. The following is a list of the peer review reports and author responses from that submission.

Round 1

Reviewer 1 Report

The paper presents an interesting and relevant topic in today’s environment. However, there is some room for improvement. There are some minor grammatical mistakes that need to be addressed.

One area that needs to be addressed is that the abstract mentions a connection to the UN SDGs, but the paper does not address them again. I would have liked to see more of a connection between the results and SDGs and HRM. Again, with HRM the results suggest that there is a connection, but it needs to be more fully developed either in the literature review and/or the practical implications.

I would like to have seen more development of the sample. The authors mention an online and spot survey but do not offer much detail of where they got the contact information for the participants. Did the get a list of emails and if so from where. I would like to see more clarification of the background of the sample. Did the authors draw from hotel employees or was it more general? It was interesting in the limitations, the authors stated that their sample was a convenient sample but it is not clear how they determined who they would sample.

In table one the authors need to clarify that the income is monthly (at least I assume it is given that the majority of their sample comes in under 10000 RMB or less than $1,400 and the average annual salary in China is over $12,400.)

Finally, in the literature review, the authors define three hypotheses but, in the results, they have 13. If they were going to expand the hypotheses in that manner, it should have been addressed in section three of the paper. I also found it suspect that all 13 of the hypotheses were supported. It is rare for a paper to have 13 hypotheses and have all of them be supported. That needs to be clarified.

Author Response

- Thank you very much and really appreciate you for the comments! We have corrected some grammatical mistakes and made some adjustment to the literature review part of the article.

- We have made some revision to the abstract and introduction parts. In terms of literature review, we added some references published in recent years. And we have made some adjustments to the discussion and implication part. As for the lay-out of hypotheses, we have added the sub-hypotheses to the main part of the article.

- Due to limitation of time, we might not make big adjustments to the methodology part shortly and we have to carefully check the original documents. According to the questionnaires we collected, we have found that all the hypotheses are supported through statistical analysis.

Reviewer 2 Report

The study investigates the relationship between employee perceptions of CSR and its influence on organizational identity and commitment. It is not very clear from the literature and background information, how this study is unique in its contribution. Similar complex studies have been undertaken, especially in the hospitality industry. The authors need to provide information regarding unique contributions of this study. Extensive English editing is required. Citation format is not according to journal guidelines. Introduction and literature review section need to be strengthened extensively with up-to-date references. Numerous citations are very old. Social identity theory is used as framework, however, the discussion does not tie back in to the theory used. The discussion section is limited to implications. Detailed discussion of the findings and related conclusions need to be drawn. Specifically, the discussions need to identify other studies that may have found similar results. I have serious concerns related to data collection method and sample. The entire manuscript seems very scattered with no specific focus. Too many concepts are introduced without context. Overall, the manuscript needs careful and detailed revision in all aspects of the study to be accepted. Specific comments are listed below:

The introduction does not provide adequate background information and is difficult to follow in numerous places. It does not provide adequate information to provide basis for the purpose statement. Social identity theory is used, however, adequate citations need to be provided. Direct quotes used do not provide page number. Purpose statement described in lines 60 – 64 is different from what is outlined in lines 98-101. The literature review seems scattered and does not specifically describe literature pertinent to the research study. Discussion related to social entrepreneur does not have context. Lines 125 – 129 describe the relationships between different variables. However, it gets confusing when the authors indicate that organizational identity is both a mediator and dependent variable. Also, this paragraph is not required in this section. The current study uses dated references. CSR studies, especially in the hospitality industry have been undertaken in recent years. There are numerous studies that have investigated relationships examined by the current study. How is the current study unique in its contributions? The analysis conducted for the study need an introduction. Authors are requested to provide the survey instrument. The lit review section does not identify different elements of CSR as measured in the instrument. Also, in the results section, the authors provide series of hypothesis (H1a,b, c…), which is not indicated in previous sections. The authors collected data using 2 methods. Analysis needs to be undertaken to ensure that there was no difference in data due to variance in data collection method. While extensive statistical analyses have been conducted, it is not clear why these were conducted. CFA results may be provided in tables with short description. EFA results may be omitted or mentioned briefly. One of the primary concerns related to the study is the sample. The sample consists of panel participants of an online survey company who may or may not be employees in the hotel/hospitality industry. It is also not clear how many participants were from the panel and how many were recruited through “spot investigation”. “Spot investigation” is also not a valid statistical method for data collection. For the findings to be meaningful, the authors will need to collect data from appropriate venues. While statistically the data provides support for the hypotheses, it may not be valid for the purpose of the study.

Author Response

- Thank you very much and really appreciate you for the comments! We have corrected some grammatical mistakes and made some adjustment to the literature review part of the article.

- We have made some revision to the abstract and introduction parts. In terms of literature review, we added some references published in recent years. And we have made some adjustments to the discussion and implication part, we’ve split and updated the Chapter 6 into Discussions and Conclusions to make it easier to understand and logical.

- As for the lay-out of hypotheses, we have added the sub-hypotheses to the main part of the article. Due to limitation of time, we might not make adjustments to the methodology part shortly and we have to carefully check the original documents.

Reviewer 3 Report

Dear Authors,

The submitted paper is interesting and the topic is current. Also, the data collected from the research are valuable to the scientific community.

The paper must be improved in the following aspects:

the Abstract should be reformulated (especially lines 23-28) - the relationship between CSR and SDGs is not discussed enough maybe the Keywords can be selected more carefully the motivation of the research should be emphasized in Introduction Chapter 2 does not have sufficient scientific support to exist as a separate chapter a new and more consistent chapter should be introduced for
Literature review. More international scholars should be addressed and more recent papers should be discussed Hypothesis 4 is not reflected in Figure 1 there is an inconsistency the age of the employees (line 282) I suggest to split and update Chapter 6 into Discussions and Conclusions, to make it easier to read the text

Best regards,

Reviewer

Author Response

- Thank you very much and really appreciate for your comments. We’ve cited more articles published in recent years in the literature review part and also added some references.

- We’ve made some changes to Figure 1 to show four hypotheses clearly, and made the age of employees more consistent in the table.

- We’ve split and updated the Chapter 6 into Discussions and Conclusions to make it easier to understand and logical.

Round 2

Reviewer 3 Report

Dear Authors,

I still believe the data you collected through your research is valuable and can generate more valuable information.

The improvements from the last version were minor and they even introduced more confusion into the text. I recommend you to reorganize the whole paper and define some clear research hypothesis that you could verify through your research.

Gook luck for future research!

Author Response

Thank you very much and really appreciate you for the comments! We have corrected some grammatical mistakes and made some adjustment to the literature review part of the article. We have made some revision to the abstract and introduction parts. We have reorganized the whole paper and defined the research hypothesis according to your instructions.
